



# Solitary wave characteristics on the fine structure of mesospheric sporadic sodium layer

*Shican Qiu[1,2]\*, Mengxi Shi[1], Willie Soon[3,4], Mingjiao Jia[5], Xianghui Xue[2,6], Tao Li[2,6], Peng Ju[1], Xiankang Dou[2,6]\**

[1] Department of Geophysics, College of the Geology Engineering and Geomatics, Chang'an University, Xi'an, 710054, China.

[2] Key Laboratory of Geospace Environment, Chinese Academy of Sciences, University of Science & Technology of China, Hefei, Anhui, 230026, China

[3] Center for Environmental Research and Earth Sciences (CERES), Salem, MA 01970, USA

[4] Institute of Earth Physics and Space Science (ELKH EPSS), 9400, Sopron, Hungary.

[5] Shandong Guoyao Quantum Lidar Co., Ltd., Jinan, 250101, China.

[6] Mengcheng National Geophysical Observatory, School of Earth and Space Sciences, University of Science and Technology of China, Hefei, Anhui, 230026, China.

*Correspondence to*: Shican Qiu (*scq@ustc.edu.cn*) and Xiankang Dou (*dou@ustc.edu.cn*)

**Abstract.** The most fantastic phenomenon of the mesospheric sodium layer is the so-called sporadic sodium layer (SSL or $Na_S$), which are proposed to be closely related to wave fluctuations. Solitary wave is a particular solution of partial differential equation whose energy travels as a localized wave packet, and a soliton is a special solitary wave which has a particle-like behavior with strong stable form. In this research, the solitary wave theory is applied for the first time to study the fine structure of $Na_S$. We perform soliton fitting processes on the observed data from the Andes Lidar Observatory, and find out that 24/27 $Na_S$ events exhibit similar features/characteristics to a soliton. Time series of the net anomaly of the $Na_S$ reveal the same variation process to the solution of a generalized five-order KdV equation. Our results therefore suggest the $Na_S$ phenomenon would be an appropriate tracer for nonlinear wave studies in the atmosphere.

Keywords: sporadic sodium layers, solitary waves, soliton, lidar



# 1 Introduction

The sodium layer locates at about 80–110 km above, normally with a Gaussian distribution formed through the ablation of meteors (Kane and Gardner, 1993; Kopp, 1997). The most fantastic phenomena of the sodium layer is the so-called sporadic sodium layer (SSL or $Na_S$) (Cox et al., 1993; Gardner et al., 1993; Mathews et al., 1993). For the $Na_S$ event, the sodium density will increase rapidly and could be more than double the background value (e.g. with an intensity factor >2) within several minutes in a narrow altitude range (Hansen and von Zahn, 1990). $Na_S$ last from tens of minutes to several hours (Nagasawa and Abo, 1995; Prasanth et al., 2007), and their full width at half maximum (FWHM) usually less than 5 km or sometimes only 1–2 km (Hansen and von Zahn, 1990; Nagasawa and Abo, 1995; Prasanth et al., 2007). Since first reported on 1978 (Clemesha et al., 1978), a lot of mechanisms have been proposed (Cox et al., 1993; von Zahn et al., 1987; Zhou et al., 1993). The current evidences conform the ion-molecule theory is the most possible mechanism for $Na_S$, based on the local observation (Dou et al., 2009; Dou et al., 2010; Hansen and von Zahn, 1990; Heinrich et al., 2008; Heinselman et al., 1998; Kirkwood and Nilsson, 2000; Nesse et al., 2008; Qiu et al., 2016; von Zahn and Hansen, 1988; Williams et al., 2006), and model simulation (Collins et al., 2002; Cox et al., 1993; Cox and Plane, 1998; Plane et al., 2015; Plane, 2003). The sodium ions from $E_S$ could probably provide sufficient neutral sodium atom source through recombination with free electrons (Collins et al., 2002; Cox et al., 1993; Cox and Plane, 1998; Plane et al., 2015; Plane, 2003; Qiu et al., 2020).

On the other hand, mesospheric sodium layer observations by lidars are believed to be a tracer to possibly identify the atmospheric wave signals (Gardner and Voelz, 1987; Gardner et al., 2019; Gong et al., 2015; Li et al., 2007a; Li et al., 2007b; Xu and Smith, 2004). The current existing mechanisms indicate that $Na_S$ is closely related to wave fluctuations (Clemesha et al., 1997; Kane et al., 1991; Qian et al., 1998; Zhou et al., 1993; Zhou and Mathews, 1995). Many observational results reveal $Na_S$ often occur accompanied by gravity waves (Ban et al., 2015; Li et al., 2007a; Li et al., 2007b; Qian et al., 1998). Meanwhile, the fine structures of $Na_S$ exhibit distinct characters related to waves on small time scale (Chen and Yi, 2011; Liu and Yi, 2009; Liu et al., 2013). The bursts of the sodium atoms show a pulse period of 30 seconds (Liu and Yi, 2009), indicating a wave fluctuation effect on the evolution of $Na_S$.

Recently, a peculiar kind of the nonlinear waves, which is called the solitary wave, has been widely studied (Belashov and Vladimirov, 2005; Benci and Fortunato, 2014; Wazwaz, 2009). It was first reported by John Scott Russell (1808–1882), when he was observing the motion of a boat rapidly drawn along a narrow channel (Russell, 1844). From then on, many researchers have made a lot of studies on the theoretical derivation and exploration on this nonlinear problem. Generally speaking, the solitary wave is a particular solution of partial differential equation whose energy travels as a localized wave packet, and a soliton is a special solitary wave which has a particle-like behavior with strong stable form (Benci and Fortunato, 2014). Soliton is of great interest to the modern physics and mathematics (Belashov and Vladimirov, 2005; Benci and Fortunato, 2014; Wazwaz, 2009). It is a fundamental wave structure for the nonlinear wave processes, which plays an important role in the wide spectrum of areas of research related to the wave physics, e.g., in hydrodynamics, plasma physics, condensed matter, and optics (Belashov and Vladimirov, 2005). The solitary waves nowadays are widely applied for the



Earth science especially involving ocean and atmosphere, such as surface waves, ion-acoustic waves, magnetosonic waves, internal gravity waves, Raleigh waves off a seismic source, and so on (Belashov and Vladimirov, 2005).

The nonlinear solitary wave theory was first decisively exerted by Korteweg and de Vries as the simplified model equation for surface waves on shallow water,

$$\frac{\partial u}{\partial t} + \alpha u \frac{\partial u}{\partial x} + \beta \frac{\partial^3 u}{\partial x^3} = 0, \tag{1}$$

with solutions of stable solitary waves [*Korteweg and Vries*, 1895](Korteweg and Vries, 1895). The term soliton was first introduced in 1965 by Zabusky and Kruskal who demonstrated that the Korteweg-de Vries equation (KdV equation) reveals hidden linear properties, allowing a solution in the form of a nonlinear solitary wave propagating without changing its profile (Zabusky and Kruskal, 1965). These authors also pointed out that the soliton has two important properties: (1) extremely stable wave packet like a particle; (2) invariant even under particle collisions (Zabusky and Kruskal, 1965).

In this research, the solitary wave theory has been deduced and compared with the lidar observations. We perform particular data processing on the observed results from a narrow band lidar at the Andes Lidar Observatory (Liu et al., 2016). We find that the fine structure of $Na_S$ evolutions exhibit similar characters to a soliton, indicating a common existence of solitary waves on the mesopause region. The evolution of the net anomaly of the $Na_S$ peak profiles exhibit the same characters to the solution of five-order KdV equation. Our results therefore suggest the $Na_S$ phenomenon would be a possible tracer for nonlinear wave studies.

## 2 The solitary wave theory and data processing

### 2.1 The nonlinear and dispersion effects

Consider a one-dimensional wave at the moment t, the particle number density at x in the medium is given by $n(x, t)$. Under the conservation of particle number, there is

$$\frac{dn}{dt} = 0, \tag{2}$$

where *n* could possibly be regarded as the input of sodium sources from $Na^+$ through chemical reactions. After the input, $n(x, t)$ would possibly undergo the evolution ways: the convergence or divergence.

The full derivative form for $n(x, t)$ can be expanded as

$$\frac{\partial n}{\partial t} + v \frac{\partial n}{\partial x} = 0, \tag{3}$$

where $\frac{dx}{\partial t} = v$ is the velocity of the particle movement. This equation has a generalized solution:

$$n(x, t) = f(x - v(n)t). \tag{4}$$

This solution indicates each part of the wave has a different speed of $v(n)$. When $\frac{dv}{dn} > 0$, $v(n)$ would increase with increasing density n. Therefore the wave packet front becomes steeper and steeper as the wave propagates, leading to a nonlinear effect of convergence (Fig. 1a).


On the other hand, a wave propagating in x direction can be expressed as

$$u(x, t) = A\exp[i(kx - \omega t)], \tag{5}$$

where u is the wave function, A is the amplitude, k is the wave number, and $\omega$ is the angular frequency. The phase velocity $v_p$ and group velocity $v_g$ are given by,

$$v_p = \frac{\omega}{k}, \tag{6}$$

and

$$v_g = \frac{\partial \omega}{\partial k}, \tag{7}$$

respectively. If $\frac{\omega}{k} \neq$ constant, each wavelet with different wave vector will have distinguished velocity. So the term $\omega = \omega(k)$ indicates a dispersion effect of the wave packet (Fig. 1b).

Particularly, the dispersion term of a surface wave in incompressible shallow fluid is given by

$$\omega(k) = \sqrt{gh}k - \frac{1}{6}\sqrt{gh}h^2 k^3, \tag{8}$$

where h is the fluid depth, g is the gravitational acceleration (Belashov and Vladimirov, 2005).

In the complex space, we have $\frac{\partial}{\partial t} \leftrightarrow -i\omega$, and $\frac{\partial}{\partial x} \leftrightarrow ik$. Substituting into to Eq. (8), we obtain

$$\frac{\partial u}{\partial t} + \left(\sqrt{gh} + u\right)\frac{\partial u}{\partial x} + \frac{1}{6}\sqrt{gh}h^2 \frac{\partial^3 u}{\partial x^3} = 0. \tag{9}$$

Set $u' = u + \sqrt{gh}$, and $\beta = \frac{1}{6}\sqrt{gh}h^2$, then the equation is given by

$$\frac{\partial u'}{\partial t} + u'\frac{\partial u'}{\partial x} + \beta \frac{\partial^3 u'}{\partial x^3} = 0. \tag{10}$$

This equation (similar to Eq. (1)) is one of the simplest forms of the KdV equation, balanced by both the nonlinear term $u\frac{\partial u}{\partial x}$ and dispersion term $\frac{\partial^3 u}{\partial x^3}$. Hereby the solution satisfying this equation will undergo no convergence or dispersion effect, and

the wave shape could be maintained for a long time.

## 2.2 The solution of KdV equation and numerical simulation

Travelling wave could be represented by the form $u(x, t) = f(x - ct)$, where $u(x, t)$ represents a disturbance moving in the negative or positive x direction if $c < 0$ or $c > 0$, respectively [*Wazwaz*, 2009](Wazwaz, 2009). If the solution $u(x, t)$

25

depends only on the difference between the two coordinates of the partial differential equations, then the solution keeps its exact shape, and therefore called solitary waves. So a solitary wave is a travelling wave whose transition from the asymptotic state at $\xi = -\infty$ to the other asymptotic state at $\xi = \infty$ is localized in $\xi$, where $\xi = x - ct$, and c is the wave speed (Wazwaz, 2009).

Equation (10) or (1) has a special solution given by





$$u(x, t) = u_2 + (u_1 - u_2)\text{sech}^2 \sqrt{\frac{u_1 - u_2}{12\beta}}(x - ct), \qquad (11)$$

where $u_1$ and $u_2$ are exhibited by Fig. 2a, and $\text{sech}$ is referred to the hyperbolic secant function (Zabusky and Kruskal, 1965). This is just the bow wave observed by Russell in the early years, e.g. a solitary wave (Zabusky and Kruskal, 1965). Then $u_2$ represents the limiting wave amplitude of Eq. (11) at infinity, and $u_1$ characterizes the peak of the wave. Set $a = u_1 - u_2$ and

$d = \sqrt{12\beta/a}$, thus $a$ is the amplitude and $d$ is the width of the wave (shown by the vertical distance between the two red dashes of Fig. 2a).  Figure 1a could possibly corroborate some descriptions of solitary wave properties as: (1) This wave propagates along x direction, e.g. with the form of $u(x - ct)$; (2) This wave is distributed in a limited space, e.g. $\lim\limits_{x \to \pm\infty} u \to 0$; (3) The shape of the wave does not change with time. This specific kind of nonlinear wave is therefore called a soliton.

**3 Observation results and discussions**

The observational data from the Andes lidar from August 20, 2014 to July 7, 2019 are processed in detail as follows: (1) Select the typical $Na_S$ event with intensity factor >3, and determine the Gaussian distribution function of the background sodium density profile on that day. (2) Subtract the Gaussian distribution from the original peak density profile of the $Na_S$. The net anomaly peak is obtained and next fitted by the soliton solution from the standard KdV equation. Find out the net anomaly distribution function and evaluate the quality of the fitting. (3) Compare evolutions of the net anomaly with the

solution of a generalized five-order KdV equation. Make a video to illustrate their variation processes, and intercept single-frames for this comparison.

**3.1 The Gaussian distribution of the sodium density profile**

It is shown that the sodium density variation with height can be approximated by the Gaussian distribution,

$$n_s(z) = \frac{C_s}{\sqrt{2\pi}\sigma_s} \exp[-\frac{(z - z_s)^2}{2\sigma_s^2}], \qquad (12)$$

where $n_s(z)$ is the sodium number density at z, $C_s$ is the total column density of the sodium layer, $z_s$ is the centroid height, and $\sigma_s^2$ is the RMS width (Xue, 2007).

When the $Na_S$ occur, the sodium density suddenly increases in a narrow altitude and the density profile obviously deviates from the Gaussian distribution. To quantitatively explore this anomaly, the Gaussian distribution function of the background sodium density should be determined first.

The sodium density data observed at one night by the lidar compose of a two-dimensional matrix: the elements on the column vectors of the matrix represent the sodium densities at different heights at a given moment, and the elements on the row vectors represent the results at different moments at a given height. Now we choose the column vector of the matrix for data processing.

Set the origin observation data matrix of the day to be $\overleftrightarrow{\mathbf{M}}$, then $\overleftrightarrow{\mathbf{M}}$ is represented by the column vector as





$$\overleftrightarrow{\mathbf{M}} = [\vec{m}_1, \vec{m}_2, \dots, \vec{m}_{n-1}, \vec{m}_n], \tag{13}$$

where n is the number of observational points on the day. Mark the maximum element in $\overleftrightarrow{\mathbf{M}}$ as $d_{max}$. If we select an $Na_S$ event with intensity factor $> 3$, the critical value $d_c$ for determining the anomalies is defined as

$$d_c = \frac{d_{max}}{3}, \tag{14}$$

when the value of an element in the column vector is greater than $d_c$, it reflects the anomaly of Na density. Otherwise, it is considered that the column vector conforms to the Gaussian distribution of Na density on that day. The column vector reflecting the Na density anomaly is arranged into a matrix in the order of the observed time as

$$\overleftrightarrow{\mathbf{A}} = [\vec{a}_1, \vec{a}_2, \dots, \vec{a}_{k-1}, \vec{a}_k]. \tag{15}$$

And then set the matrix of column vectors matching the Gaussian distribution in the order of observational moments to be

$$\overleftrightarrow{\mathbf{G}} = [\vec{g}_1, \vec{g}_2, \dots, \vec{g}_{n-k-1}, \vec{g}_{n-k}] \tag{16}$$

To obtain the Gaussian distribution of Na density on that day, we average all column vectors in $\overleftrightarrow{\mathbf{G}}$ with a mark of $\vec{g}_{ave}$:

$$\vec{g}_{ave} = \frac{\sum_{i=1}^{n-k} \vec{g}_i}{n-k}, \tag{17}$$

where $\vec{g}_{ave}$ reflects the distribution of Na density at altitude. As mentioned above, it is considered that $\vec{g}_{ave}$ is consistent with the Gaussian distribution, i.e., the result of a Gaussian fit with $\vec{g}_{ave}$ to the altitude h should satisfy Eq. (12).

Taking the $Na_S$ observed on November 3, 2016 for example, the Gaussian fitting is made with the column vector $\vec{g}_{ave}$ and $\vec{h}$ according to Eq. (12). The fitting results are as follows

$$\vec{g}_h = 4330 \exp[-\frac{(\vec{h}-92.86)^2}{7.798}], \tag{18}$$

where $\vec{g}_h$ represents the modeled value that corresponds to the Gaussian distribution. The fitting result is shown as the red curve in Fig. 2c.

## 3.2 The net anomaly of sodium density and solitary wave fitting

Based on the observational data at Andes station from 20 August 2014 to 7 July 2019, 27 $Na_S$ events are distinguished among 147 observation days. Continue to take the case on November 3, 2016 as an example. The column vector of $d_{max}$ is denoted as $\vec{a}_d$ ($1 \le d \le k$). The profile of $\vec{a}_d$ with $\vec{h}$ is shown as the blue dash-dot line in Fig. 2c. The observed Na density can be regarded as the sum of Gaussian distribution and the anomaly. Thus, the vector $\vec{p}$ reflecting the net anomaly of Na density is given by the following equation

$$\vec{p} = \vec{a}_d - \vec{g}_h, \tag{19}$$

and the distribution of $\vec{p}$ with $\vec{h}$ is shown as the blue dotted line in Fig. 2d. Then the peak density of $|\vec{p}|$ equals to 10180.95 $cm^{-3}$, with $|\vec{p}|$ to about zero at infinity.



Then the solitary wave fitting is performed on the vector $\vec{p}$. Since $\vec{p}$ reflects the net anomaly at a determined time (the moment of the peak density profile), the parameter t in the fitted formula is a constant. At this time the traveling wave $\xi = x - ct$ represents a specific phase. In order to better represent the altitude corresponding to the peak Na density, Eq. (11) could be written as

$$u(\xi) = u_2 + (u_1 - u_2)\operatorname{sech}^2 \sqrt{\frac{u_1 - u_2}{12\beta}}(\xi - \xi_0), \tag{20}$$

where $\xi_0$ is the altitude corresponding to the maximum amplitude of the solitary wave. The fitting expression is finally deduced as

$$u(\xi) = -82.17 + 10263.12 * \operatorname{sech}^2 \sqrt{\frac{10263.12}{19494.36}}(\xi - 93.63), \tag{21}$$

which means

$$u_2 = -82.17,$$
$$u_1 - u_2 = 10263.12 \text{ (or } u_1 = 10180.95),$$
$$12\beta = 19494.36,$$
$$\text{and}$$
$$\xi_0 = 93.63.$$

The fitting results are shown as the red curve in Fig. 2d. Now we know $u_1$ represents the peak density of the modeled curve, and $u_2$ corresponds to the density at infinity, i.e.

$$u(\xi) \underset{\xi \leq 80}{\approx} u(\xi) \underset{\xi \geq 110}{\approx} \lim_{\xi \to \infty} u(\xi) \to 0 \tag{22}$$

Let

$$\xi_1 = \pm\frac{d}{2} = \pm\sqrt{\frac{3\beta}{a}} = \pm\sqrt{3\beta/(u_1 - u_2)}, \tag{23}$$

putting it to Eq. (21), we have

$$u(\xi_1) = u_2 + \frac{4e}{(e+1)^2}(u_1 - u_2) = u(\pm 0.6891) = 8071.41, \tag{24}$$

which means the density at 0.6891 km from the peak is predicted to be 8071.41 ($cm^{-3}$). From Fig. 2d, we can find all these simulated values, $u_1$, $u_2$ and $u(\xi_1)$ are close to the observed results.

Furthermore, the theoretical full width at half maximum (FWHM) of a solution is given as:

$$h_{d/2} = \xi_0 + \frac{d}{2} = \xi_0 + \frac{\sqrt{\frac{12\beta}{u_1 - u_2}}}{2} \ (km). \tag{25}$$

To test the fitting results, the observed altitude of the selected case needs to be close to the calculated $h_{0.5}$. Since the vertical resolution of the lidar is limited to 0.5 km, we can perform linear interpolation for the observed altitude series. The interpolated vector $\vec{p}$ is denoted as $\vec{p_1}$, and the altitude series of $\vec{p_1}$ is marked as $\vec{h_1}$. The Na density at half width is $u(h_{d/2})$, then it is always possible to find a value closest to $u(h_{d/2})$ on both sides of the peak of $\vec{p_1}$ (denoted as $p_1(m)$ and $p_1(n)$).





The corresponding altitudes for $p_1(m)$ and $p_1(n)$ are noted as $h_1(m)$ and $h_1(n)$, respectively. Then the observed width of the soliton could be given as:

$$d' = h_1(n) - h_1(m). \tag{26}$$

According to the definition of the soliton, $d'$ is also written as follows:

$$d' = \sqrt{\frac{12\beta'}{\max(\overline{p_1}) - 0}}, \tag{27}$$

where $\max(\overrightarrow{p_1})$ is the peak of $\overrightarrow{p_1}$;

0 is the density at infinity;

and $\beta'$ is the $\beta$ value deduced from the observation. It is shown that

$$\beta' = \frac{d'^2 \max(\overline{p_1})}{12}. \tag{28}$$

The deduced values of $h_{d/2}$, $d$, $d'$, $u(h_{d/2})$, $p_1$, $\beta$ and $\beta'$, for each Na$_S$ event, are listed in Table 1. The fitting parameters on November 3, 2016 are also exhibited on Fig. 2d. Then the results show the fitting parameters are close to the observed values.

On the other hand, the column vector obtained by Eq. (21) is denoted as $\vec{q}$, then $\vec{q}$ is the predicted value at altitude h. Then some parameters to evaluate the fitting quality are needed. The coefficient of determination ($|R^2| < 1$) essentially measures the residual sum. The root mean square (RMSE) is also called the fitting standard deviation of a regression. They are calculated, respectively, as

$$R^2 = 1 - \frac{\sum_{i=1}^{L}(q_i - p_i)^2}{\sum_{i=1}^{L}(p_i - \overline{p}_i)^2}, \tag{29}$$

and

$$RMSE = \sqrt{\frac{1}{L}\sum_{i=1}^{L}(q_i - p_i)^2}, \tag{30}$$

where L is the length of vector $\vec{p}$ or $\vec{q}$, i.e., the number of observed altitude points.

The denominator of Eq. (29) denotes the residuals obtained by predicting net anomaly vector $\vec{p}$, and the numerator represents the residuals predicted using the modeled vector $\vec{q}$. When $R^2 < 0$, it means that the residuals of the results predicted by the anomaly model are larger than those obtained from the mean value of the vector $\vec{p}$ reflecting the net anomaly, indicating a poor result. When $R^2 > 0$, the larger $R^2$ indicates that the residuals are smaller, and the predicted effect is better.

The values of the parameters $R^2$ and RMSE obtained by fitting $\vec{p}$ and $\vec{q}$ for this Na$_S$ event are calculated, respectively, as

$$R^2 = 0.9784,$$

and

$$RMSE = 315.8402,$$



showing a good fit with $R^2$ close to 1. Through the evaluation of fitting quality, it is confirmed that Eq. (23) can fit the net anomaly vector $\vec{p}$ within an allowable error. The fitting results and quality evaluation parameters are obtained, with 24/27 events having $R^2 > 0.9$ as shown in table 1. Therefore, the fitting results indicate $u(\xi)$ are consistent with the net anomaly $|\vec{p}|$.

### 3.3 Further explanation of Na$_S$ by higher-order solitary wave equation

In Fig. 2d, it is shown that some small wavelets appear on both wings of the blue dotted line drawn from the net anomaly vector $\vec{p}$, which could not be explained by the standard solitary equation with the red fitting curve. These wavelets, however, have similar characteristics to the higher-order solitary wave with a waveform shown as Fig. 2b.

Kawahara and Takuji proposed to add a higher-order dispersion term to the KdV equation, which considers dissipation, instability and higher-order dispersion effect in medium (Kawahara and Takuji, 2007). This generalized KdV equation is written as

$$\frac{\partial u}{\partial t} + u\frac{\partial u}{\partial x} + \beta\frac{\partial^3 u}{\partial x^3} + \gamma\frac{\partial^5 u}{\partial x^5} = 0, \tag{31}$$

which is also called the Kawahara equation (Kawahara and Takuji, 2007).

The numerical simulation results show that the Kawahara equation has two types of solutions: in the case of $\gamma > 0$ and $\beta \leq 0$, a soliton is formed with monotonic asymptotics similar to the soliton shown in Fig. 2a; in the case of $\gamma > 0$ and $\beta > 0$, the two wings of the soliton will have oscillation characteristics [*Mamun and Shukla*, 2009] (Mamun and Shukla, 2009). Taking $\beta = 1$ and $\gamma = 10^{-4}$, i.e., the expression of equation (28) will be

$$\frac{\partial u}{\partial t} + u\frac{\partial u}{\partial x} + \frac{\partial^3 u}{\partial x^3} + 10^{-4}\frac{\partial^5 u}{\partial x^5} = 0, \tag{32}$$

whose solution at $t = 0.0016$ is shown as Fig. 1b.

Further, we can study the evolution of the waveform over time using the Fourier transform method. Let the initial conditions be $\beta = 0.0023, \gamma = 10^{-4}$, and $u(x, 0) = e^{-36x^2}$. Since the Kawahara equation contains two variables, x and t, the simulation results are denoted as a two-dimensional matrix $\overleftrightarrow{W}$. Then arrange $\overleftrightarrow{W}$ by column vectors as follows

$$\overleftrightarrow{W} = [\vec{w}_1, \vec{w}_2, \ldots, \vec{w}_{n-1}, \vec{w}_n]. \tag{33}$$

Since the vertical scale of the Na$_S$ is defined less than 10 km, while the horizontal scale is often reported more than 300 km or even over 1000 km (Fan et al., 2007; Ma et al., 2019), this scenario is in consistent with the shallow water model. The solitary waves could Particularly appear at the interface between upper and lower stratifications in the fluid medium [*Bogucki and Garrett*, 1993](Bogucki and Garrett, 1993). They are frequently found in the stratified or sheared places in the ocean and atmosphere (Doviak et al., 1991; Gan and Ingram, 1992; Huthnance, 1989). Therefore, it is more effective to look

for observational results related to stable stratifications or shears. The Na$_S$ event on April 9, 2019 is selected as an example (Fig. 3a). This Na$_S$ appears before the beginning of the observation at about 95 km altitude, with a downward propagation.



The deduced vertical wind confirms the $Na_S$ occurs in downward vertical wind (shown as the negative value region in Fig. 3b). Figure 3c – e represent the zonal wind, meridional wind, and the temperature profiles, respectively. These three profiles indicate the $Na_S$ locating near the stratification with strong shear. Further, the stability of the stratification is determined by the Richardson number (Ri) (Liu, 2011; Nappo, 2002). Under stable stratification, both convection and turbulence are less likely to develop.

In unit time, due to vertical displacement, for a unit mass of air the convective flow energy to resist the net Archimedes buoyancy is

$$W_1 = -\overline{\frac{d}{dt}\left(\frac{1}{2}w^2\right)} = K_H N^2,\qquad(34)$$

where $\qquad\qquad\qquad\qquad$ $w = dz/dt$, the vertical speed;

$\qquad\qquad\qquad\qquad$ $K_H = \overline{wz} > 0$, the convective conductivity coefficient;

$\qquad\qquad\qquad\qquad$ $N^2 = \frac{g}{T}\left(\frac{\partial T}{\partial z} + \frac{g}{C_p}\right)$, the buoyancy frequency;

$\qquad\qquad\qquad\qquad$ $g = 9.5\ ms^{-2}$, the gravitational acceleration in the mesopause;

$\qquad\qquad\qquad\qquad$ $C_p = 1004\ JK^{-1}kg^{-1}$, the specific heat at constant pressure;

and T is the atmospheric thermodynamic temperature.

On the other hand, for unit mass of air, the horizontal kinetic energy consumed per unit time in the presence of vertical wind shear, i.e., the kinetic energy provided to convective motion, is

$$W_2 = -\overline{\frac{d}{dt}\left(\frac{1}{2}\vec{V}_h^2\right)} = K_M\left[\left(\frac{\partial u}{\partial z}\right)^2 + \left(\frac{\partial v}{\partial z}\right)^2\right],\qquad(35)$$

where u and v are the zonal and meridional wind velocities, respectively. $K_M > 0$ is the momentum transport coefficient and is usually approximately equal to $K_H$.

Thus, the stability of the layer depends on the value of $W_1/W_2$. When $W_2$ is less than $W_1$, it means that the disturbance kinetic energy converted by the basic airflow is less than the disturbance kinetic energy consumed by the stable stratification. In this situation, even if convection or turbulence occurs, it will be suppressed or weakened, so the atmosphere is in a stable state. The dimensionless ratio, $W_1/W_2$, is defined as

$$Ri = \frac{W_1}{W_2} = \frac{\frac{g}{T}\left(\frac{\partial T}{\partial z} + \frac{g}{C_p}\right)}{\left[\left(\frac{\partial u}{\partial z}\right)^2 + \left(\frac{\partial v}{\partial z}\right)^2\right]},\qquad(36)$$

which could be deduced from the wind and temperature results observed by the lidar (Fig. 3c – e). The calculated Ri with a value >1.5 are shown as black scatters in Fig. 3f. The $Na_S$ locates around the area where scatters are concentrated, i.e., a special stable area near 95 km. It is obvious that the $Na_S$ and stable region evolved synchronously, and finally both become blurred. Therefore, the lidar observations and deduced results are all consistent with the appearance of a solitary wave. A stable stratification with Ri >1.5 exists near about 95 km, accompanied by the zonal wind, meridional wind and temperature





being also stratified around 95 km. Once a fluctuation is excited in the vicinity of the stratification and just satisfies the balance of nonlinear and dispersion effects, the waveform will maintain through the propagation, i.e., a solitary wave appears.

Furthermore, the evolution of observed net anomaly for the $Na_s$ throughout whole night is also needed for a comparison. In section 3.1, the initial observation of sodium density is denoted as

$$\overleftrightarrow{M} = [\vec{m}_1, \vec{m}_2, ..., \vec{m}_{n-1}, \vec{m}_n].$$

Note the Gaussian distribution model of the current day as a two-dimensional matrix $\overleftrightarrow{G_0}$, and define $\overleftrightarrow{G_0}$ as isomorphic to $\overleftrightarrow{M}$. Then $\overleftrightarrow{G_0}$ could be written in the form of column vector as

$$\overleftrightarrow{G_0} = [\vec{g}_h, \vec{g}_h, ..., \vec{g}_h, \vec{g}_h]. \tag{37}$$

Obviously, the net anomaly evolution of the $Na_S$ is obtained by observation data matrix $\overleftrightarrow{M}$ subtracting the Gaussian

distribution model $\overleftrightarrow{G_0}$, which could be denoted as $\overleftrightarrow{S}$. There are

$$\overleftrightarrow{S} = \overleftrightarrow{M} - \overleftrightarrow{G_0} = [\vec{m}_1 - \vec{g}_h, \vec{m}_2 - \vec{g}_h, ..., \vec{m}_{n-1} - \vec{g}_h, \vec{m}_n - \vec{g}_h]$$
$$= [\vec{s}_1, \vec{s}_2, ..., \vec{s}_{n-1}, \vec{s}_n], \tag{38}$$

where $\vec{s}_i (i = 1、2、...、n - 1、n)$ is the column vector of $\overleftrightarrow{S}$.

In order to compare the numerical simulation results $\overleftrightarrow{W}$ with the $Na_S$ evolution model $\overleftrightarrow{S}$ more intuitively, the Fourier

transform method is used uniformly to disperse a certain period of time on the time variable t into n moments in the numerical simulation. Thus, both $\overleftrightarrow{W}$ and $\overleftrightarrow{S}$ are two-dimensional matrices with the same column number n. To better show them, a dynamic video of the variation of their column vectors with time has been made (uploaded as Data Repository, https://dx.doi.org/10.12176/01.99.02129). The five left images in Fig. 4 (i.e., Fig. 3a, c, e, g, i) are single-frame intercepted from $\overleftrightarrow{W}$ of the video, showing the evolutions of the five-order solitary wave over time. And the five images on the right (i.e.,

Fig. 4b, d, f, h, j) are intercepted from $\overleftrightarrow{S}$, indicating variations of the net anomaly of the $Na_S$. Figure 4a and b show at this moment, the simulated wave shape are similar to the observed peak density profile. Figure 4c and d: the huge peaks attenuate gradually. Figure 4e and f: the peaks decay to about zero value. Figure 4g and h: the peaks change phase and resume. Figure 4i and j: the peaks recover to a sharp form similar to the initial condition, except with different phase. By comparison, it is verified that the numerical simulation results $\overleftrightarrow{W}$ are in good agreement with the evolution process $\overleftrightarrow{S}$ of the $Na_S$. So the five-

25
order solitary wave theory is appropriate in explaining this $Na_S$ event.

However, it is worth noting that the numerical simulation of the higher-order KdV equation is probably only suitable for explaining the events similar to the selected case. These events are typically characterized by occurrence heights below 95 km, longer durations, and descending patterns similar to tidal fluctuations. In contrast, the other events with shorter durations and cloud-like shapes are less consistent with the higher-order simulation results. This discrepancy also implies

30
that the $Na_S$ with different characteristics may have different fine structures.



**4 Conclusions**

In this research, the solitary wave theory is applied to study the Na$_S$. Among the observations of Andes lidar from August 20, 2014 to July 7, 2019, 27 Na$_S$ cases with intensity factor>3 have been selected for processing through the Gaussian and soliton fitting steps. The original observed peak density profile of the Na$_S$ is subtracted by the Gaussian distribution, and then the net anomaly peak is obtained. The net peak is fitted by the soliton solution from the standard KdV equation, and the quality of the fitting is then evaluated. Statistical results reveal that in 24/27 cases, the net peak of Na$_S$ exhibits similar features to a soliton. Time series of the net anomaly on 9 April, 2019 reveals a similar variation process to the solution of a five-order KdV equation. Although still highly uncertain, this solitary wave theory could possibly explain some characteristics of Na$_S$.

**Data availability**

The regular ALO Na Lidar data between 80 and 105 km is publicly available from the Andes Lidar Observatory database at http://lidar.erau.edu/data/nalidar/.

**Acknowledgements**

This work is supported by the National Natural Science Foundation of China (NO. 41974178 and 41831071), CNSA pre-research Project on Civil Aerospace Technologies (NO. D020105). We acknowledge the use of data from the Andes Lidar Observatory database. The authors express sincere respect to Prof. Alan Liu from Center for Space and Atmospheric Research & Department of Physical Sciences, Embry-Riddle Aeronautical University, USA, for providing the valuable data.

**Author information**

**Affiliations**

**Department of Geophysics, College of the Geology Engineering and Geomatics, Chang'an University, Xi'an, 710054, China**

Shican Qiu, Mengxi Shi, & Peng Ju

**Key Laboratory of Geospace Environment, Chinese Academy of Sciences, University of Science & Technology of China, Hefei, Anhui, 230026**

Shican Qiu, Xianghui Xue, Tao li & Xiankang Dou

**Center for Environmental Research and Earth Sciences (CERES), Salem, MA 01970, USA**

Willie Soon

**Institute of Earth Physics and Space Science (ELKH EPSS), 9400, Sopron, Hungary**

Willie Soon

**Shandong Guoyao Quantum Lidar Co., Ltd., Jinan, 250101, China**




Mingjiao Jia

**Mengcheng National Geophysical Observatory, School of Earth and Space Sciences, University of Science and Technology of China, Hefei, Anhui, 230026, China**

Xianghui Xue, Tao li & Xiankang Dou

**Contributions**

Shican Qiu conceived this study and wrote this manuscript.

Mengxi Shi performed data analysis and prepared Fig. 1 and Fig. 3.

Willie Soon was in charge of the organization and English polishing of the whole manuscript.

Mingjiao Jia prepared Fig. 2 and gave some useful comments on the content.

Xianghui Xue wrote the response to reviewers and added some materials in the discussion.

Tao Li helped with the response to reviewers.

Peng Ju helped discuss with the KdV equations.

Xiankang Dou conceived this study and provided data from the Chinese Meridian Project.

**Competing interests**

The authors declare no conflict of interest.

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





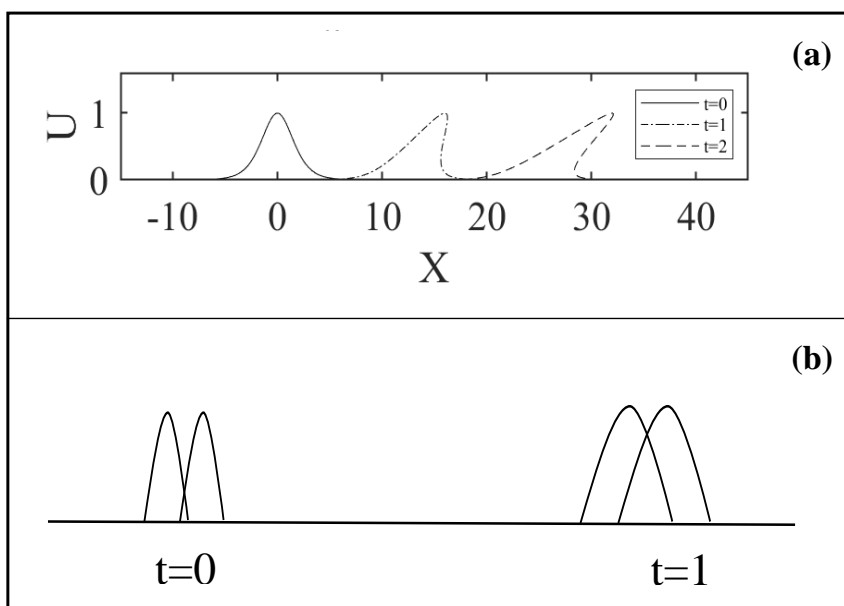

Figure 1: The convergence (a) and dispersion (b) effects of a wave. (a) The wave packet front becomes steeper and steeper as the wave propagates, leading to a nonlinear effect of convergence. (b) The dispersion effect for the wave packet.



**Figure 2: Comparison of the theoretical solitary wave profiles and observed peak density profiles of Na$_S$. (a) The solitary wave profile according to Eq. (11), with u$_2$ representing the limiting wave amplitude at infinity and u$_1$ characterizing the peak of the wave. The width d $= \sqrt{12\beta/a}$ is determined by the vertical distance between the two red dashes. (b) Simulated solution of the five-order solitary wave. (c) The peak density profile of the Na$_S$. The blue dash-dot line represents the peak sodium density observed on November 03, 2016. The red curve indicates fitted background Gaussian distribution throughout the whole night. (d) The fitting image of the peak profile. The blue dotted line shows the distribution of the observed data after subtracting the background Gaussian distribution from 2c, while the red curve is simulated according to Eq. (11) with appropriate parameters. The blue dotted line is very close to the red curve, except some wing features similar to 2b. The fitting parameters of $h_{d/2}, d, d', u(h_{d/2})$ and $p_1(n)$ are also given.**





Figure 3: Observations and results from the Andes lidar on April 9, 2019. The empty areas indicate a low signal-to-noise ratio (SNR) and large error of the observed data. (a) The sodium density profile. The Nas appears before the beginning of the observation, at about 95 km altitude. (b) The vertical wind observations. (c) The zonal wind profile. (d) The meridional wind profile. (e) The temperature variations. (f) The calculated Ri distributions. The black scatter dots represent Ri with a value >1.5.







**Figure 4: Comparison of five-order solitary wave evolution images over time and Na density net anomalies observed at the Andes station on April 9, 2019. (a) and (b): at this moment, the simulated wave shape is similar to the observed peak density profile. (c) and (d): the huge peaks attenuate synchronously. (e) and (f): the peaks decay to about zero value. g and h: the peaks change phase and resume. (i) and (j): the peaks recover to a sharp form similar to the initial condition, except with different phase. A dynamic video of the variation of their column vectors with time has been made and uploaded as DOI:10.12176/01.99.02129.**





**Table 1. Statistics of fitting parameters and fitting quality evaluation for the observations at Andes station from August 20, 2014 to July 7, 2019**

| Parameter \ Event | Parameter of Solitary Equation | | | | | | Parameter of Solitary Wave Width | | | | | |
|---|---|---|---|---|---|---|---|---|---|---|---|---|
| | $u_1$ $(cm^{-3})$ | $u_2$ $(cm^{-3})$ | $\beta$ $(m^{-1})$ $\times 10^{12}$ | $\xi_0$ (km) | RMSE | $R^2$ | $h_{d/2}$ (km) | $d$ (km) | $d'$ (km) | $u(h_{d/2})$ $(cm^{-3})$ | $p_1(n)$ $(cm^{-3})$ | $\beta'$ $(m^{-1})$ $\times 10^{12}$ |
| 2014-08-20 | 12441.3 | -102.3 | 877.0 | 90.46 | 462.0 | 0.9612 | 90.92 | 0.92 | 0.85 | 9761.5 | 9723.6 | 763.3 |
| 2015-01-30 | 21676.5 | -185.9 | 5938.7 | 94.06 | 625.9 | 0.9867 | 94.96 | 1.81 | 2 | 16982.8 | 16965.3 | 6997.9 |
| 2015-02-02 | 42591.4 | 638.8 | 6625.1 | 94.56 | 2226.9 | 0.945 | 95.25 | 1.38 | 1.3 | 33585.2 | 33714.0 | 6100.0 |
| 2015-04-18 | 14265.7 | -221.2 | 6860.8 | 98.03 | 1462.5 | 0.8814 | 99.22 | 2.38 | 1.45 | 11170.3 | 11090.7 | 2624.0 |
| 2015-04-19 | 12100.9 | -294.4 | 10571.7 | 89.2 | 1151.0 | 0.9152 | 90.80 | 3.20 | 2.8 | 9460.0 | 9425.1 | 9028.6 |
| 2015-04-21 | 10863.3 | -699.4 | 4055.2 | 91.58 | 643.8 | 0.9561 | 92.61 | 2.05 | 2 | 8381.8 | 8423.7 | 3547.1 |
| 2015-04-22 | 15853.3 | 494.6 | 3668.7 | 88.95 | 894.3 | 0.9445 | 89.80 | 1.69 | 1.55 | 12564.1 | 12637.9 | 3406.4 |
| 2015-11-06 | 12555.5 | 16.5 | 755.3 | 92.55 | 532.6 | 0.9457 | 92.98 | 0.85 | 0.85 | 9857.4 | 9803.9 | 731.7 |
| 2016-02-25 | 5602.8 | 158.2 | 1312.0 | 91.99 | 277.0 | 0.9573 | 92.84 | 1.70 | 1.7 | 4451.5 | 4445.6 | 1329.6 |
| 2016-03-02 | 7023.3 | 107.5 | 378.2 | 92.96 | 238.1 | 0.962 | 93.37 | 0.81 | 0.7 | 5565.7 | 5536.8 | 294.7 |
| 2016-03-15 | 6590.4 | 325.7 | 2703.1 | 94.94 | 491.5 | 0.9224 | 96.08 | 2.28 | 2.3 | 5255.4 | 5260.4 | 2980.0 |
| 2016-06-06 | 10644.9 | 586.4 | 2464.4 | 91.23 | 634.5 | 0.9362 | 92.09 | 1.71 | 1.75 | 8498.7 | 8426.6 | 2876.3 |
| 2016-10-26 | 16359.1 | -252.6 | 1197.3 | 96.88 | 1076.1 | 0.8772 | 97.35 | 0.93 | 0.85 | 12863.6 | 12744.1 | 962.2 |
| 2016-10-28 | 7372.1 | 12.1 | 776.0 | 97.33 | 412.2 | 0.9192 | 97.89 | 1.12 | 1.1 | 5796.4 | 5810.6 | 720.3 |
| 2016-11-03 | 10181.0 | -82.2 | 1624.5 | 93.63 | 315.8 | 0.9784 | 94.32 | 1.38 | 1.35 | 7979.4 | 7914.6 | 1536.4 |
| 2016-11-09 | 12595.5 | 82.1 | 753.6 | 92.8 | 620.9 | 0.9181 | 93.23 | 0.85 | 0.8 | 9966.6 | 9880.7 | 638.1 |
| 2017-04-22 | 31064.8 | -424.9 | 3944.1 | 96.29 | 1298.5 | 0.9578 | 96.90 | 1.23 | 1.25 | 24363.0 | 24502.0 | 4016.3 |
| 2017-11-25 | 9205.2 | -31.2 | 699.7 | 94.79 | 256.4 | 0.9784 | 95.27 | 0.95 | 0.9 | 7245.7 | 7158.5 | 600.1 |
| 2017-11-28 | 8995.4 | -8.7 | 974.1 | 97.05 | 454.1 | 0.9416 | 97.62 | 1.14 | 1.15 | 7085.2 | 7066.9 | 978.4 |
| 2017-12-16 | 11306.7 | 193.5 | 2482.5 | 96.48 | 656.9 | 0.9414 | 97.30 | 1.64 | 1.5 | 8928.1 | 8945.1 | 2249.8 |
| 2017-12-17 | 7019.4 | 1.6 | 828.7 | 92.64 | 159.1 | 0.9881 | 93.24 | 1.19 | 1.2 | 5526.0 | 5525.2 | 814.2 |
| 2017-12-19 | 12824.9 | -68.1 | 1388.5 | 96.1 | 493.6 | 0.9654 | 96.67 | 1.14 | 5.8 | 2977.6 | 2995.4 | 18319.9 |
| 2017-12-21 | 4522.8 | -20.5 | 4225.3 | 94.44 | 786.3 | 0.7611 | 96.11 | 3.34 | 2.95 | 5423.3 | 5435.3 | 5842.3 |
| 2017-12-22 | 7491.7 | -61.9 | 3933.2 | 94.66 | 696.0 | 0.9024 | 95.91 | 2.50 | 1.3 | 8190.9 | 8208.6 | 2131.7 |
| 2019-04-07 | 17210.2 | 66.4 | 641.5 | 95.29 | 403.8 | 0.9784 | 95.63 | 0.67 | 1.9 | 13379.5 | 13314.1 | 5090.4 |
| 2019-04-09 | 18618.4 | 488.4 | 2708.2 | 94.82 | 1178.4 | 0.9179 | 95.49 | 1.34 | 1.45 | 14755.6 | 14738.4 | 2964.7 |
| 2019-07-06 | 12787.4 | -256.7 | 3345.4 | 92.97 | 546.2 | 0.9714 | 93.85 | 1.75 | 1.65 | 10020.7 | 9985.5 | 2934.3 |