# Peer review of "Solitary wave characteristics on the fine structure of mesospheric sporadic sodium layer"

_Atmospheric Chemistry and Physics, 2021_

## Referee Comment (RC2)

The authors apply a solitary wave model to numerous sporadic Na layer ($Na_s$) profiles measured with a lidar at the Andes Lidar Observatory and find that for most events, a solitary wave model provides very good fits to the $Na_s$ profiles. The implication is that these $Na_s$ are linked to and may be somehow caused by the solitary wave. The paper is adequately referenced but I found the writing quite confusing in places. Section 2, where the fundamental solitary wave theory is discussed needs a major rewrite as do Section 3.1, 3.2 and 3.3 where the fitting of the theory to data is discussed. I understand what they are doing, but it was figure 2, not the text that enabled me to figure things out. Even so I'm puzzled by some of the equations, for example, should equation (11) be written as

$$u(x,t) = u_2 + (u_1 - u_2)sech^2\left[\sqrt{\frac{u_1-u_2}{12\beta}}(x-ct)\right] ?$$

Although the idea that $Na_s$ may be related to solitary waves is interesting, my main concern is that the authors have provided no insight into how the Na density could rise to such large values in $Na_s$ simply by the passage of a solitary wave through the Na layer. The conventional explanation for $Na_s$ is that very high concentrations $Na^+$ are collected in thin layers by the combined effects of the earth's magnetic field and the vertical wind shears caused by large amplitude waves and tides. Chemical reactions then convert the Na ions to neutral Na, thus forming the $Na_s$. While the authors have demonstrated that the solitary wave model provides a good fit to the $Na_s$, this is hardly evidence that solitary waves are involved. $Na_s$ are thin, sometimes form rapidly, and often show vertical phase progression that mimic the phase progression of long period waves and tides. The authors do not discuss those issues. At a minimum, they need to show how a solitary wave propagating through the mesopause region would impact the density profile of minor species like Na. Such theoretical work has been done for waves and tides (e.g. Gardner & Shelton, JGR, 90(A2), pp. 1745-1754, 1985), but not for solitary waves, which behave differently.

I recommend that the paper be returned to the authors for major revisions, that address the issues I have raised. I hope they do so because if they can show how a solitary wave produces the thin $Na_s$ with vertical phase progression, like that illustrated in Figure 3a, then this would provide convincing evidence that solitary waves are frequent in the mesopause region and deserve more attention from the upper atmosphere research community.

---

## Author Comment (AC1)

We would like to pay special thanks to the reviewer for valuable comments and constructive suggestions. We took a closer look at all the comments and reviewed the manuscript accordingly. All changes in red fonts have been marked in the revised manuscript. The explicit answers to the comments are given below in blue fonts.

Review "Solitary wave characteristics on the fine structure of mesospheric

sporadic sodium layer" by Qiu et al.

In the current version of the manuscript, Qiu et al. have reported their work on studies of the sporadic sodium layer (Nas), one of the most active research areas in the upper atmosphere subject. They have introduced this subject concisely in the abstract section, for example, the definition of Nas and the possible mechansim (e.g., ion-molecule chemistry mechanism and recombination with electrons) as well as Nas relationship with waves. Then the authors have introduced the solitary wave theory and has applied it to try to explain the fine structure of mesospheric Nas observed by a narrow band lidar at Andes Lidar Observatory.

1. In section 2, the authors have described the solitary wave theory but they have assumed that the particle density changes with time is constant, which is probably wrong in particular for the sporadic Na layer for this study. Acturally I am quite struggled to understand the sections 2 and 3 but it looks that using the the solitary wave fitting method clearly matches the observation data better than after they have selected Nas cases. Based on this and their Figure 4, the authors then conclude that "this solitary wave theory could possibly explain some characteristics of Nas".

Thanks for the worthy comment. Gardner and Shelton 1985 in previous studies made the following hypothesis about the formation of  $Na_S$  by gravitational waves: The velocity field of the minor constituent (e.g. sodium) equals to the atmospheric velocity field. Only wave-induced dynamics are considered; no chemical effects associated with atomic sodium are part of these solutions. This has the effect of reducing the source and loss terms, P and Q in Eq. (1\*\*\*), to zero. Eq. (1\*\*\*) is written as:

$$\frac{\partial n}{\partial t} + \nabla \cdot (n\mathbf{V}) = P - Q = 0 \quad , \qquad (1^{***})$$

where

 $n \sim$  density of the minor constituent;  $V \sim$  velocity field of the minor constituent;  $P \sim$  source terms;

and

 $Q \sim loss$  terms.

In our selected event, solitary wave occurred where both horizontal and vertical winds shear strongly, which means that V in Eq.  $(1^{***})$  is also equal to zero.

Thanks again for the reviewer's brilliant insight. As a result, we can now choose the NaS situations using the solitary wave fitting approach.

2. The authors have also warned that "it is worth noting that the numerical simulation of the higher-order KdV equation is probably only suitable for explaining the events similar to the selected case" and "In contrast, the other events with shorter durations and cloud-like shapes are less consistent with the higher-order simulation results. This discrepancy also implies 30 that the NaS with different characteristics may have different fine structures." This is quite confusing and there is not clear conclusions and convincing results from the current work. It also depends on the method the authors have applied to do the data analysis (for example, they have applied the gaussian fit first for the lidar measurements dataset then subtract it then use the anomlay to do the analysis, see their method in Page 5).

Thanks for the kind and worthy comment. Russell discovered a solitary wave that moves ahead in shallow, narrow channels and maintains its form and speed for an extended period of time. In reality, similar theoretical investigations reveal that solitary wave propagation is relatively stable. Further, theoretical studies show that the propagation of solitary waves is relatively steady. This implies that, as Russell proposes, singular waves tend to travel for a long time. Based on this, we propose the opinion that shorter duration, cloud-like events with poor agreements to higher-order simulations. The results cannot be explained by isolated wave theory, which imply that NaS with different characteristics may have different fine structures.

3. It looks the guassian fit used in the Equation 12 is not suitable for the Nas layer (Shown in the Figure 2c). Is that the reason to apply the solitary fitting for the density aomaly? If so, the caption (measureed data) in the Figure 2d is misleading because it is the Na density difference from Lidar and guassian fit. If you choose to different Guassian fit (for example, super gaussian fit function), will the result be different?

Thanks for the kind comment. During the handling of the raw data, we according to the size of the density value determined the boundary of an exception occurrence and if is not an exception occurs, then select all, there was no abnormal data on highly sequence gauss fitting, fitting results show as Figure 2c red solid line, blue dotted line in Figure 2c said in peak time of the original Na density data. On the other hand, we have modified the legend in Figure 2d according to your suggestion. The mathematical expression of super-Gaussian fitting can be written as:

$$f(x) = \exp(-(\frac{x}{w})^N), N \ge 2.$$

W is a constant. If n is 2, this expression is reduced to a Gaussian distribution, and if n is an even number greater than 2, it is a super-Gaussian distribution. The following figure shows the image when w is 20 and n is 2, 6, and 10 respectively.

Figure 1\*\*\*

Apparently, when there is no Nas layer, the super-Gaussian fitting is more inconsistent with the actual observed density distribution of Na layer.

4. Somehow, I am lost in understanding how to obtain the Eq. (21). The parameters used in the fitting expression are different for different cases (which shown in Table 4). So my feeling is that these parameter will give better fit for the data rather than an explaination of Nas layer.

Thanks for your question. Eq. (36) has been obtained from Eq. (35) and  $\xi_0$  (the height of maximum Na density observed by lidar). In other words, by substituting the value of  $\xi_0$  into Eq. (35) and using the least square fitting method to match the data  $\vec{p}$ , Eq. (36) can be obtained.

5. There are also many assumptions and it is very hard to judge and is unclear if they are reasonable or not. For example, For the equation (2), why Na "could possible be regarded as the input of sodium sources from Na+ through chemical reactions"? Please keep in mind the source of Na is from the the ablation of incoming meteors.

Thanks for the kind feedback. Propelled by your review opinion above, we think that the expression of the variable 'n' could be regarded as  $Na^+$  produced. And from some previous references, we know the input of sodium could also be the meteor injection (although the amount of the injection could hardly explain the huge increase of sodium density). As the kind reviewer 1 suggests the input could also be controlled by dynamic processes, we have added the three sources for 'n'. We made some changes to the statements in the revised manuscript (page 3, line 27).

6. Again, why the authors assume the same airmasses (conservation of particle number)

in Equation (2) to let dn/dt equals zero? This mean the production and loss term of Na is always the same. Is this applicable for Nas layer?

Thanks for your question. In previous studies (Gardner and Shelton 1985) on the formation of NaS by gravitational waves, the following hypotheses were made: The velocity field of the minor constituent (e.g. sodium) equals the atmospheric velocity field. Only wave-induced dynamics are considered; no chemical effects associated with atomic sodium are included in these solutions. This has the effect of reducing the Source and loss terms, P and Q in Eq.  $(2^{***})$ , to zero. Eq.  $(2^{***})$  is written as:

$$\frac{\partial n}{\partial t} + \nabla \cdot (n\mathbf{V}) = P - Q = 0 \quad , \qquad (2^{***})$$

where

$$\label{eq:velocity} \begin{split} n &\sim density \ of \ the \ minor \ constituent; \\ V &\sim velocity \ field \ of \ the \ minor \ constituent; \\ P &\sim source \ terms; \end{split}$$

and

 $Q \sim loss$  terms.

In our selected event, solitary wave occurs where both horizontal and vertical winds shear strongly, which means that V in Eq.  $(2^{***})$  is also equal to zero. That is the Eq.  $(2^{***})$  in our manuscript.

7. Why the authors only consider "the dispersion term of a surface wave in incompressible shallow fluid" In equation (8) and how this equation 8) is derived? Does this mean that only one single Nas layer can be done in the current method? However, from the Lidar observations, it looks that the peak Nas layer occurrs at different altitude (here just shows one case 2015-02-02 used in the Table 1, see the figure at

http://lidar.erau.edu/data/nalidar/plots/2015/20150202\_Dmerge\_15min\_0.5km\_90s\_2 0\_p.jpg)

Thanks for your question. With Bernoulli's equation for ideal fluid :

$$\frac{\partial u}{\partial t} = -\frac{1}{\rho} \frac{\partial p'}{\partial x}$$

$$\frac{\partial w}{\partial t} = -\frac{1}{\rho} \frac{\partial p'}{\partial z}$$

$$\frac{\partial u}{\partial x} + \frac{\partial w}{\partial z} = 0$$
(3\*\*\*)

Its boundary conditions are:

$$(w|_{z=0} = 0, (\frac{\partial p'}{\partial t} - \rho g w)|_{z=h} = 0.$$
 (4\*\*\*)

Where u represents the horizontal velocity and w represents the vertical velocity, p' is pressure.

Let  $\mathcal{L} \equiv \frac{\partial}{\partial t} \left( \frac{\partial^2}{\partial x^2} + \frac{\partial^2}{\partial z^2} \right)$ Eq. (3\*\*\*) is written as

$$\mathcal{L}\mathbf{w} = \mathbf{0} \tag{5***}$$

Let

$$w = W(z)e^{i(kx-\omega t)}, \qquad (6^{***})$$

Substitute into equation  $(5^{***})$

$$\frac{d^2W}{dz^2} - k^2 W = 0 \tag{7***}$$

consequently,

$$W(z) = Ae^{kz} + Be^{-kz}$$
(8\*\*\*)

where A and B are constants.

Eq. (8\*\*\*) Substitute into Eq. (5\*\*\*), get:

$$v = (Ae^{kz} + Be^{-kz})e^{i(kx-\omega t)}$$
(9\*\*\*)

Then, put Eq. (9\*\*\*) into Eq. (3\*\*\*)

$$p' = \frac{i\omega\rho}{k} (Ae^{kz} - Be^{-kz})e^{i(kx-\omega t)}, \qquad (10^{***})$$

According to the lower boundary conditions :

$$B = -A; \tag{11***}$$

According to the upper boundary conditions :

$$\{(\frac{\omega^2}{k} - g)e^{kh} - (\frac{\omega^2}{k} + g)e^{-kh}\}A = 0.$$
 (12\*\*\*)

So

$$\omega = \sqrt{gk \tanh(kh)}.$$
 (13\*\*\*)

In shallow water conditions,

$$\omega = \sqrt{gk[kh - \frac{1}{3}(kh)^3]} \approx \sqrt{k^2 c_0^2 (1 - \frac{1}{3}k^2 H^2)}$$

=  $kc_0 (1 - \frac{1}{6}k^2 H^2) = kc_0 - \frac{1}{6}k^3 c_0 H^2.$  (14\*\*\*)

Eq.  $(14^{***})$  is the dispersion relation.

H. Ikezi et al. 's research (Ikezi et al., 1970) on solitons shows that they have properties similar to particle collisions, an apparent linear interaction (two peaks temporarily merging into one larger peak) is observed when the two solitons pass through each other from opposite directions. Observations of Na atoms in the mesosphere also frequently show this characteristic, and our hypothesis is that the Nas duration in a relatively short period of time may be the result of the interaction of two or more solitons. Since lidar observations are made only at a certain location, it is unrealistic to describe the possible collision process from the available data. The following figure shows several consecutive Nas events that may support our view.

8. What is the value of "the fluid depth h" used for different cases because this is required to calculate the depth of wave d?

Thanks for the valuable comment. Grimshaw et al. pointed out that different nonlinear evolution equations derived by applying different boundary conditions in atmospheric media, and KdV equation can be derived when the boundary conditions are set to rigid cover (Grimshaw, 2002). Crook believed that for any real atmospheric region with long wave and enough drastic stability changes, it could be regarded as the rigid cap hypothesis, that is, the KdV equation was derived (crook, 1988; crook, 1986). However, due to the limitation of observation conditions, we cannot give the numerical value of the fluid layer depth, but it is natural to infer that the isolated waves observed should be related to the boundary between the stable and unstable regions of the atmosphere. Fortunately, the stability of atmospheric regions can be reflected by Richardson Numbers, which may help us to derive some information about the depth of the fluid layer.

9. My other major concern is the lack of the explaination for sporadic Na Layer formation in the current version of the manuscript, which seems to me it still unclear why solitary wave causes the sporadic Na layer. If we look at one case used in the current manuscript, for example their Figure 3, there is strong correlation of Nas layer (Figure 3a) with zonal mean wind shear in Figure3f where Richardson number is calculated from the Equation 36, which has the zonal mean wind changes with altitude from the lidar data. Of course, this suggests that atmospheric waves are related to the Nas formation, but how the authors can attribute it to solitary waves, instead of gravity waves or tidal waves etc. This may be not true because the authors have not applied a

similar analysis for the the neutral Na data without Nas layer. If the result using the neutral Na data by ignoring Nas is similar as presented in the current manuscript, then that would indicate the current conclusion is wrong. To be specified, what the results will look like if the authors apply the same method to the neutral Na data excluding Nas layer? I understand that may be tough since there are some creteria to be met (for example, set the Na concentration and layer depth).

Thanks for the comment. Absolutely this paper does not hope to invoke a new mechanism for the source of the Nas, it just focuses on the time series of fine structure. The authors always believe that the source of sodium atoms comes from the ion-molecule reactions or dynamic effects. The results from our data processes indicate Nas could be a possible tracer for nonlinear wave studies through its time series. The fine structure of Nas exhibits distinct wave fluctuations (shown as Figure 4 in the revised manuscript) and wing-like features (shown as Figure 2d). And as we know the C structure billow-like feature of Nas (shown as the following image) could be hardly explained by only the source of sodium atoms. As a result, the fine structure of the Nas could be a possible tracer to study the wave fluctuation and atmospheric instability.

Figure 3\*\*\*

---

## Author Comment (AC2)

We would like to pay special thanks to the reviewer for valuable comments and constructive suggestions. We took a closer look at all the comments and reviewed the manuscript accordingly. All changes in red fonts have been marked in the revised manuscript. The explicit answers to the comments are given below in blue fonts.

Sporadic sodium layer events are frequently observed by the Na lidars around the world. This study provides another potential underline mechanism that could be helpful for the understanding of this dramatic event in the upper atmosphere. Here, the author utilizes the solitary wave theory to explain several sporadic sodium layer events in the upper mesosphere observed by a Na Doppler lidar at ALO. The waveform of the solidary wave is derived by fitting the residual of the observed Na layer anomalies to the solution of the KdV equation that describes the solitary wave. The author suggests that the observed "solidary wave" is "consistent with the shallow water model", and presents an example of $Na_S$ coexisting with a strong wind shear in the lidar data. I know ALO has several nightglow instruments operating as well, so is it possible to check and see if these nightglow instrument captured any of these reported "solidary wave" events in Table 1?

Thank you so much for sharing your opinion. We tried our best to support our results with the observational data. Unfortunately, we were unable to obtain Nightglow Imager Observational data from the ALO station. Additionally, in order to further test our theory, it's required to obtain the data of $Na_S$ propagation velocity but with the current technical conditions there, acquisition is quite difficult. It would have been interesting to explore this aspect and support our claim through it however, presents situation lags advancement in this regard.

This would strengthen the author's argument with further experimental evidence. In addition, why does not the author fit the $5^{th}$ order KdV solution for all of these event, since it appears the $5^{th}$ order solution generates better fitting?

Thanks for your kind comments and sharing reservations. The grounds for introducing 5th order KdV is to explain the tiny wavelets that appear on both wings of the blue dotted line in Figure 2d in the revised manuscript. As stated in the Eq. (46), once the higher order dispersion factor is included into the KdV equation, wavelets begin to appear on both sides along the main peaks of the solitary wave solution and it is more consistent with the fine structure of $Na_S$ observed. Further, we concentrated on the role of dispersion rather than the sequence of dispersion in defining the fine structure of $Na_S$.

On the other hand, my major concern is the author seems to suggest the $Na_S$ is the product of $Na^+$ layer in solidary wave form through Na ion-molecular chemistry, if I understand it correctly. A sharp $Na^+$ layer with high peak ion density near and below 90 km, where neutral-molecular chemistry dominates, is highly unlikely in my

opinion. There is not argument of the chemical reaction time, and how it compares with time of the wave event. In addition, the author completely ignores the possibility that it can also be a dynamic feature.

Technical comments:

1.  The title of the paper does not reflect the key point of the manuscript. Since the whole paper is focusing on solidary wave mechanism for Na$_S$, it would be beneficial to somehow include "solidary wave" in the title.

    Thanks for the constructive suggestions. As per your valuable suggestion, we have revised the title of the manuscript as: Solitary wave characteristics on the fine structure of mesospheric sporadic sodium layer.

2.  Page 2, Line 2: The author states the Na layer shape is "normally with a Gaussian distribution". But This is really depending on the temporal resolution of how the lidar data are processed. For short time scale, the layer does not appear to be Gaussian at all. If the "solidary wave" lasted less than one hour in the Na lidar observations, this assumption will not apply. The author should be very careful about this statement. So more clarification would be required.

    Thanks for your feedback. We cordially appreciate your concern and totally agree with your opinion that for short time scales, this layer doesn't appear to be Gaussian at all. However, none of the 27 samples we picked had a time range of fewer than seven hours in real data processing. (Table 1* elaborates the starting and ending time of each case)

    In addition, for the case of November 3, 2016, we plotted the density profile at numerous periods other than the peak moment (as shown in Figure 1*). Their corresponding moments are five, two and one hour before the peak chronologically and one, two and five hours after the peak. The red curve represents a fitted background Gaussian distribution throughout the night.

[Figure]

[Figure]

[Figure]

Figure 1*

Table 1*

| Event | observation start time (UT) | observation end time (UT) | Duration (hour) |
|---|---|---|---|
| 2014-08-20 | 0.5 | 10.2 | 9.7 |
| 2015-01-30 | 0.9 | 9 | 8.1 |
| 2015-02-02 | 1.6 | 9 | 7.4 |
| 2015-04-18 | -0.5 | 9.7 | 10.2 |
| 2015-04-19 | -0.9 | 9.5 | 10.4 |
| 2015-04-21 | -0.3 | 9.8 | 10.1 |
| 2015-04-22 | -0.3 | 9.8 | 10.1 |
| 2015-11-06 | -0.2 | 8.8 | 9 |
| 2016-02-25 | 1.1 | 9 | 7.9 |
| 2016-03-02 | -0.2 | 8.8 | 9 |
| 2016-03-15 | 0.1 | 8.6 | 8.5 |
| 2016-06-06 | 1.1 | 10.6 | 9.5 |
| 2016-10-26 | -0.5 | 8.8 | 9.3 |
| 2016-10-28 | -0.1 | 8.3 | 8.4 |
| 2016-11-03 | 0 | 8.8 | 8.8 |
| 2016-11-09 | 0.1 | 7.8 | 7.7 |
| 2017-04-22 | -0.4 | 9.8 | 10.2 |
| 2017-11-25 | 0.6 | 8.5 | 7.9 |
| 2017-11-28 | 0.1 | 7.9 | 7.8 |
| 2017-12-16 | 0.7 | 8.9 | 8.2 |
| 2017-12-17 | 0.2 | 8.9 | 8.7 |
| 2017-12-19 | 1.9 | 8.9 | 7 |
| 2017-12-21 | 0.6 | 9 | 8.4 |

| 2017-12-22 | 0.3 | 8.9 | 8.6 |
|---|---|---|---|
| 2019-04-07 | -0.4 | 10.2 | 10.6 |
| 2019-04-09 | -0.8 | 10.1 | 10.9 |
| 2019-07-06 | 0.9 | 10.9 | 10 |

3. Page 2, Line 10: The author states "the ion-molecular theory is the most possible mechanism for $Na_S$". This statement is still debatable. Although there is high correlation between $Na_S$ and Es (or $Na^+$) in the MLT, the dynamic effected cannot be ruled out, since the ion-neutral collision rate is still high, especially near and below ~100 km. In fact, some recent simulations indicate the dynamic effect can play important role in the Na layer structure in the lower thermosphere up to ~120 km or even higher.

Thanks for your suggestion. We totally agree with you. Gardner et al. derived the relationship between $Na_S$ and gravitational waves in details, which was supported by many subsequent observations. We have corrected the imprecise error statements and added corresponding references (Page 2, line 10).

4. Page 2, Line 24-25: Most of these references were published more than a decade ago, so I would not say they are "recently".

Thanks for your comment. We have replaced the de facto expression "Recently" with "In the last decade and earlier" (page 2, line 27).

5. Page 3, line 14: should be "in the mesopause region".

Thanks for the kind suggestion. We cordially apologize for our error and this typo has been corrected (page 3, line 19).

6. Page 3, after equation 2, the author states the variable 'n' could be regarded as $Na^+$ produced Na. I understand this statement follows the previous one (#3). But, again, I think the author should also consider/include the possibility of dynamic transport of Na atoms in the argument.

Thanks for your valuable comment. We apologize for the manuscript's fundamental objective being misunderstood. Our research paper has no intention of invoking a new mechanism for the $Na_S$ source. It is primarily

concerned about the time series of fine structure. Influenced by your 3rd review opinion above, the expression of the variable 'n' could be regarded as a production from $Na^+ \rightarrow Na$. According to Xu and Smith, 2003, it appears that the sodium density could also be concentrated through dynamic processes. And the wisdom reviewer 3 proposed the input could be meteor injection, too. Thus, the input 'n' could possibly be generated through molecule reactions, dynamics, or meteor injection. We made some changes to the statements in the revised manuscript (page 3, line 27).

7. Page 5, line 4: It reads somewhat awkward that u2 is defined before the definition of u1. In addition, what is "the limiting wave amplitude"? Please clarify.

  Thanks for the valuable comment. There are numerous sorts of solitons within the solitary waves and one common feature of them is that the amplitude of the soliton at infinity is a definite constant (Zabusky and Kruskal, 1965), which is called "the limiting amplitude". In our study, this constant is $u_2$.

8. Page 9, line 25-26: This statement need further clarification. The vertical scale of $Na_S$ is less than 10 km because of the limitation set up the Na lidar range, but it does not mean the vertical propagation of the "solidary wave" is limited to the same scale. It would be highly possible that the wave propagates beyond the mesospheric Na layer (the Na lidar range) with larger vertical scale.

  Thank you for your advice. We have no objection to your opinion and totally agree with you. The solitary wave theory has been utilized to explain well observed phenomena in the lower atmosphere and in rotating and magnetized dusty plasma in the dayside tropical mesosphere. It is therefore reasonable to suspect that solitary waves may be widespread in the earth's atmosphere. We revised the relevant statements and added corresponding references in the manuscript.

9. Page 13 on the author contribution. I do not see any data from the Chinese Meridian Project in this study, but the author Xiankang Dou "provided data from the Chinese Meridian Project"?

  Please accept our sincere apologies. Xiankang Dou provided data from the Chinese Meridian Project and conceived this study. However, in

subsequent studies, we found that there was a large error in sodium density data detected by meridian Project (as shown in the Figure 2*), and this data was not chosen for the final draft.

[Figure]

Figure 2*

10. Page 20 figure 3: The author might need to adjust the color level of the contour plots of horizontal winds, since there are some large chunks of blank area in the two plots, where the lidar data should be still good.

Thanks for your suggestion, we re-examined the raw data from ALO observations. Nothing was discovered to be lacking in the depiction of horizontal wind data.

Table 1, Would it be possible to generate a few figures to make some of the important parameters more visible, in addition to this table? It is difficult to digest the information from this busy table.

Thanks for the commentary. In order to express the information more directly, we showed the parameters $h_{d/2}$, $d$, $d'$, $u(h_{d/2})$ and $p_1(n)$ of the case on November 3, 2016 in Figure 2d.

**Cited References for this Reply:**

Xu, J., and Smith, A. K.: Perturbations of the sodium layer: controlled by chemistry or dynamics?, Geophysical Research Letters, 30, 10.1029/2003gl018040, 2003.

Zabusky, N. J., and Kruskal, M.: Interaction of "Solitons" in a Collisionless Plasma and the Recurrence of Initial States, Physical Review Letters, 15, 240–243, 10.1103/PhysRevLett.15.240, 1965.

---

## Author Comment (AC3)

We would like to pay special thanks to the reviewer for valuable comments and constructive suggestions. We took a closer look at all the comments and reviewed the manuscript accordingly. All changes in red fonts have been marked in the revised manuscript. The explicit answers to the comments are given below in blue fonts.

The authors apply a solitary wave model to numerous sporadic Na layer (Nas) profiles measured with a lidar at the Andes Lidar Observatory and find that for most events, a solitary wave model provides very good fits to the Nas profiles. The implication is that these Nas are linked to and may be somehow caused by the solitary wave. The paper is adequately referenced but I found the writing quite confusing in places. Section 2, where the fundamental solitary wave theory is discussed needs a major rewrite as do Section 3.1, 3.2 and 3.3 where the fitting of the theory to data is discussed. I understand what they are doing, but it was figure 2, not the text that enabled me to figure things out. Even so I'm puzzled by some of the equations, for example, should equation (11) be written as

$$u(x,t) = u_2 + (u_1 - u_2)sech^2\left[\sqrt{\frac{u_1-u_2}{12\beta}}(x-ct)\right] ?$$

Although the idea that Nas may be related to solitary waves is interesting, my main concern is that the authors have provided no insight into how the Na density could rise to such large values in Nas simply by the passage of a solitary wave through the Na layer. The conventional explanation for Nas is that very high concentrations $Na+$ are collected in thin layers by the combined effects of the earth's magnetic field and the vertical wind shears caused by large amplitude waves and tides. Chemical reactions then convert the Na ions to neutral Na, thus forming the Nas. While the authors have demonstrated that the solitary wave model provides a good fit to the Nas, this is hardly evidence that solitary waves are involved. Nas are thin, sometimes form rapidly, and often show vertical phase progression that mimic the phase progression of long period waves and tides. The authors do not discuss those issues. At a minimum, they need to show how a solitary wave propagating through the mesopause region would impact the density profile of minor species like Na. Such theoretical work has been done for waves and tides (e.g. Gardner & Shelton, JGR, 90(A2), pp. 1745-1754, 1985), but not for solitary waves, which behave differently.

I recommend that the paper be returned to the authors for major revisions, that address the issues I have raised. I hope they do so because if they can show how a solitary wave produces the thin Nas with vertical phase progression, like that illustrated in Figure 3a, then this would provide convincing evidence that solitary waves are frequent in the mesopause region and deserve more attention from the upper atmosphere research community.

Thanks for the comment. Starting with the Bernoulli's equation for ideal fluid:

$$\frac{\partial u}{\partial t} = -\frac{1}{\rho}\frac{\partial p'}{\partial x}$$
$$\frac{\partial w}{\partial t} = -\frac{1}{\rho}\frac{\partial p'}{\partial z} \tag{1**}$$
$$\frac{\partial u}{\partial x} + \frac{\partial w}{\partial z} = 0$$

Its boundary conditions are:

$$(w|_{z=0} = 0, (\frac{\partial p'}{\partial t} - \rho g w)|_{z=h} = 0. \tag{2**}$$

Where $u$ represents the horizontal velocity and $w$ represents the vertical velocity, p' is pressure.

Let $\quad \mathcal{L} \equiv \frac{\partial}{\partial t}(\frac{\partial^2}{\partial x^2} + \frac{\partial^2}{\partial z^2})$ ,

then the formula of Eq. (1**) is written as

$$\mathcal{L}w = 0 \tag{3**}$$

Let

$$w = W(z)e^{i(kx-\omega t)}, \tag{4**}$$

and substitute into Eq. (3**)

$$\frac{d^2W}{dz^2} - k^2W = 0 \ . \tag{5**}$$

Consequently,

$$W(z) = Ae^{kz} + Be^{-kz} \ , \tag{6**}$$

where A and B are constants.

Substitute Eq. (6**) into Eq. (3**), we have:

$$w = (Ae^{kz} + Be^{-kz})e^{i(kx-\omega t)} \tag{7**}$$

Then, put Eq. (7**) into Eq. (1**):

$$p' = \frac{i\omega\rho}{k}(Ae^{kz} - Be^{-kz})e^{i(kx-\omega t)}, \tag{8**}$$

According to the lower boundary conditions:

$$B = -A; \tag{9**}$$

According to the upper boundary conditions:

$$\{(\frac{\omega^2}{k} - g)e^{kh} - (\frac{\omega^2}{k} + g)e^{-kh}\}A = 0. \tag{10**}$$

So

$$\omega = \sqrt{gk\tanh(kh)}. \tag{11**}$$

In shallow water conditions,

$$\omega = \sqrt{gk[kh - \frac{1}{3}(kh)^3]} \approx \sqrt{k^2 c_0^2 (1 - \frac{1}{3}k^2 H^2)}$$
$$= kc_0(1 - \frac{1}{6}k^2 H^2) = kc_0 - \frac{1}{6}k^3 c_0 H^2 \tag{12**}$$

where ω is a real number. Then phase velocity $v_p$ and group velocity $v_p$ can be obtained respectively:

$$v_p = \frac{\omega}{k} = c_0 - \beta k^2, v_g = \frac{\partial \omega}{\partial k} = c_0 - 3\beta k^2 \tag{13**}$$

where

$$\beta = c_0 H^2 / 6 . \tag{14**}$$

When β≠0, $v_p \neq v_g$, which fully indicates that the $\beta \frac{\partial^3 u}{\partial x^3}$ term of KdV equation characterizes the dispersion effect.

In addition,

$$\frac{dv_g}{dk} = -6\beta k . \tag{15**}$$

Therefore, the effect of $\beta \frac{\partial^3 u}{\partial x^3}$ causes wave dispersion, With the increase of β, the wavelength becomes shorter, and the wave dispersion becomes stronger, such waves are known as dispersion waves. Of course, for long waves (when k is small), it is a weakly dispersive wave, characterized by ω containing only the odd degree term of k.

According to (Gardner and Shelton 1985), since the layer density response is highly dependent on the density gradients occurring in the layer, the steady state layer density profile becomes much important. Large density gradients encourage nonlinearities in the layer response. Therefore, when the nonlinear effect contrived by the gradient of the Na density profile is balanced with the dispersion effect mentioned above, the wave shows neither dispersion nor nonlinear characteristics, but propagates in the form of solitary waves described in the manuscript.

Atmospheric solitary wave is a kind of nonlinear gravity internal wave which is balanced between nonlinear effect and horizontal linear dispersion (Grimshaw, 2002). According to Gardner et al. (Gardner and Shelton 1985), In the quarter period

after the maximum vertical wind, the atmospheric density disturbance is the largest, and the layer peak reaches its maximum upward displacement. This has the effect of enhancing the density of secondary components above the steady-state position of the layer peak. We believe that Gardner's view can still explain the relationship between the wind field and the occurrence of the maximum of the minor components. Unfortunately, due to the lack of wind field observation data at the corresponding time, we could not reconfirm Gardner's conclusion when he discussed the linear layer response using gravity wave theory.

**Cited References for this Reply:**

Gardner, C. S., and Shelton, J. D.: Density response of neutral atmospheric layers to gravity wave perturbations, Journal of Geophysical Research, 90, 10.1029/JA090iA02p01745, 1985.

Grimshaw, R.: Environmental Stratified Flows: TOPICS IN ENVIRONMENTAL FLUID MECHANICS, THE KLUWER INTERNATIONAL SERIES, edited by: Chatwin, D. P., Dagan, D. G., List, D. J., Mei, D. C., and Savage, D. S., Springer, 285 pp., 2002.